# Classification of topological insulators and superconductors with multiple order-two point group symmetries

## Ken Shiozaki

Center for Gravitational Physics and Quantum Information,
Yukawa Institute for Theoretical Physics, Kyoto University,
Kyoto 606-8502, Japan

## Abstract

We present a method for computing the classification groups of topological insulators and superconductors in the presence of $\mathbb{Z}_2^{\times n}$ point group symmetries, for arbitrary natural numbers $n$. Each symmetry class is characterized by four possible additional symmetry types for each generator of $\mathbb{Z}_2^{\times n}$, together with bit values encoding whether pairs of generators commute or anticommute. We show that the classification is fully determined by the number of momentum- and real-space variables flipped by each generator, as well as the number of variables simultaneously flipped by any pair of generators. As a concrete illustration, we provide the complete classification table for the case of $\mathbb{Z}_2^{\times 2}$ point group symmetry.

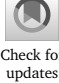
# 1  Introduction

The classification of topological insulators and superconductors was initially understood in terms of internal symmetries such as time-reversal and particle–hole symmetries [1–3]. It was later extended by incorporating crystalline symmetries [4–6], leading to the establishment of the concept of higher-order topological insulators and superconductors, characterized by boundary gapless states localized not on surfaces or edges but at corners or hinges [7–13]. These developments have greatly broadened the understanding of symmetry-protected topological phases.

In this direction, the $K$-theoretic framework developed by Freed and Moore [14, 15] has provided a general setting for the classification of topological insulators and superconductors with arbitrary crystalline symmetries. As a computational tool for $K$-theory, the Atiyah–Hirzebruch spectral sequence has been refined into a powerful and systematic method for deriving classifications [16–18]. On the other hand, for topological insulators and superconductors protected solely by (magnetic) point group symmetries without lattice translations, Cornfeld and Chapman introduced the on-site reduction of point group symmetries [19], which, together with subsequent developments [20], established a concrete method to compute the classification groups.

In this paper, we pursue this comprehensive line of study and focus on the classification of topological insulators and superconductors with multiple $\mathbb{Z}_2$ point group symmetries. While the case of a single $\mathbb{Z}_2$ point group symmetry has already been understood [21], a systematic understanding of the situation with several simultaneous $\mathbb{Z}_2$ symmetries has been lacking. Although general computational methods for classification groups with arbitrary point group symmetries have been established [19, 20], our approach provides a way to determine the relevant $K$-groups explicitly in terms of a small number of parameters. This offers practical utility, for example, in systematic classification calculations of higher-order topological insulators and superconductors under $\mathbb{Z}_2$ point group symmetries [22]. Specifically, by employing suspension isomorphisms in $K$-theory [23], we reduce the problem to the classification groups in zero dimension, thereby revealing the hierarchical structure of the classification with arbitrary numbers of $\mathbb{Z}_2$ point group symmetries. As a concrete example, we explicitly compute and present the classification table for the case of $\mathbb{Z}_2^{\times 2}$ point group symmetry.

# 2  Formulation

## 2.1  Notation for symmetries and $K$-groups

We first compute how the $K$-group in the presence of the real Altland-Zirnbauer (AZ) class and $n$ additional unitary $\mathbb{Z}_2$ symmetries can be reduced to the corresponding zero-dimensional $K$-group. As the full parameter space, we consider momentum-type variables $\boldsymbol{k} = (k_1, \ldots, k_d)$ and real-space variables surrounding a codimension $(D-1)$-defect $\boldsymbol{r} = (r_1, \ldots, r_D)$, and compactify $\mathbb{R}^{d+D}$ into the $(d+D)$-dimensional sphere $S^{d+D}$.

The eight real AZ classes are labeled by an integer $s \in \{0, \ldots, 7\}$, and we introduce the binary representation $s = (s_2 s_1 s_0)_2$, i.e.,

$$s = 4s_2 + 2s_1 + s_0, \quad s_0, s_1, s_2 \in \{0, 1\}. \tag{1}$$

The Hamiltonian $H(\mathbf{k}, \mathbf{r})$ has the following symmetries depending on the AZ class:

$$s_0 = 0 \text{ (non-chiral class)}: \quad AH(\mathbf{k}, \mathbf{r})A^{-1} = (-1)^{s_1} H(-\mathbf{k}, \mathbf{r}), \quad A^2 = (-1)^{s_2}, \tag{2}$$

$$s_0 = 1 \text{ (chiral class)}: \quad \begin{cases} AH(\mathbf{k}, \mathbf{r})A^{-1} = (-1)^{s_1} H(-\mathbf{k}, \mathbf{r}), \quad A^2 = (-1)^{s_2}, \\ \Gamma H(\mathbf{k}, \mathbf{r})\Gamma^{-1} = -H(\mathbf{k}, \mathbf{r}), \quad \Gamma^2 = 1, \\ A\Gamma = (-1)^{s_1} \Gamma A. \end{cases} \tag{3}$$

Here $\Gamma$ and $A$ denote unitary and antiunitary operators, respectively.

In addition, we consider unitary $\mathbb{Z}_2^{\times n}$ symmetries $U_1, \ldots, U_n$. According to the result of [21], each $\mathbb{Z}_2$ symmetry corresponds to four independent additional symmetry classes $t_i \in \{0, 1, 2, 3\}$ for $i = 1, \ldots, n$. For each $t_i$, we use the binary representation $t_i = (t_{i1} t_{i0})_2$, i.e.,

$$t_i = 2t_{i1} + t_{i0}, \quad t_{i0}, t_{i1} \in \{0, 1\}, \quad i = 1, \ldots, n. \tag{4}$$

Depending on the additional symmetry class $t_i$, the following relations hold [21]:

$$\begin{cases} U_i H(\mathbf{k}, \mathbf{r}) U_i^{-1} = (-1)^{t_{i0}} H(P_i \mathbf{k}, Q_i \mathbf{r}), \quad U_i^2 = 1, \\ AU_i = (-1)^{t_{i1} + (1 - s_1) t_{i0}} U_i A, \\ \Gamma U_i = (-1)^{t_{i0}} U_i \Gamma \quad \text{(when } s_0 = 1\text{)}. \end{cases} \tag{5}$$

Here $P_i$ and $Q_i$ represent diagonal $\mathbb{Z}_2$ actions on the variables $\mathbf{k}$ and $\mathbf{r}$, respectively, defined by bit strings $p_i = (p_{i1}, \ldots, p_{id}) \in \{0, 1\}^{\times d}$ and $q_i = (q_{i1}, \ldots, q_{iD}) \in \{0, 1\}^{\times D}$:

$$P_i \mathbf{k} = ((-1)^{p_{i1}} k_1, \ldots, (-1)^{p_{id}} k_d), \tag{6}$$

$$Q_i \mathbf{r} = ((-1)^{q_{i1}} r_1, \ldots, (-1)^{q_{iD}} r_D). \tag{7}$$

See Table 1 for the correspondence with the notation used in [21], which describes algebraic relations of additional symmetries. Within the sets of variables $\mathbf{k}$ and $\mathbf{r}$, we introduce the numbers of variables simultaneously flipped by $U_i$:

$$d_{i_1 \cdots i_r} = \sum_{\mu=1}^{d} p_{i_1} \cdots p_{i_r}, \quad 1 \le i_1 < \cdots < i_r \le n, \tag{8}$$

$$D_{i_1 \cdots i_r} = \sum_{\mu=1}^{D} q_{i_1} \cdots q_{i_r}, \quad 1 \le i_1 < \cdots < i_r \le n, \tag{9}$$

for $r = 1, \ldots, n$. The algebraic relations among $U_i$ are labeled as

$$U_i U_j = (-1)^{u_{ij}} U_j U_i, \quad u_{ij} \in \{0, 1\}. \tag{10}$$

With these variables, the $K$-group can be expressed as

$$K_{\mathbb{R}+nU}\Big(s, \{t_i\}_i, \{u_{ij}\}_{i<j}; d, \{d_i\}_i, \{d_{ij}\}_{i<j}, \{d_{ijk}\}_{i<j<k}, \ldots;$$
$$D, \{D_i\}_i, \{D_{ij}\}_{i<j}, \{D_{ijk}\}_{i<j<k}, \ldots\Big), \tag{11}$$

where $\mathbb{R}$ denotes the real AZ class and $nU$ indicates the presence of $n$ additional unitary $\mathbb{Z}_2$ symmetries. Following [23], we then construct the suspension isomorphism of $K$-theory, which raises the dimension of the sphere by one.

Table 1: The relation between the real AZ classes, the additional symmetry classes, and the corresponding symmetry operators. The notation follows [21]. Here $U$ denotes a symmetry commuting with the Hamiltonian, while $\overline{U}$ denotes an antisymmetry anticommuting with the Hamiltonian. The superscript of $U$ indicates the sign of $U^2 = \pm 1$. For non-chiral classes ($s \equiv 0 \bmod 2$), the subscript indicates the commutation relation with TRS or PHS. For chiral classes ($s \equiv 1 \bmod 2$), the first and second subscripts indicate commutation/anticommutation with TRS and PHS, respectively.

| AZ class | $s \backslash t_i$ | 0 | 1 | 2 | 3 |
|---|---|---|---|---|---|
| AI | 0 | $U_+^+$ | $\overline{U}_-^+$ | $U_-^+$ | $\overline{U}_+^+$ |
| BDI | 1 | $U_{++}^+$ | $\overline{U}_{-+}^+$ | $U_{--}^+$ | $\overline{U}_{+-}^+$ |
| D | 2 | $U_+^+$ | $\overline{U}_+^+$ | $U_-^+$ | $\overline{U}_-^+$ |
| DIII | 3 | $U_{++}^+$ | $\overline{U}_{-+}^+$ | $U_{--}^+$ | $\overline{U}_{+-}^+$ |
| AII | 4 | $U_+^+$ | $\overline{U}_-^+$ | $U_-^+$ | $\overline{U}_+^+$ |
| CII | 5 | $U_{++}^+$ | $\overline{U}_{-+}^+$ | $U_{--}^+$ | $\overline{U}_{+-}^+$ |
| C | 6 | $U_+^+$ | $\overline{U}_+^+$ | $U_-^+$ | $\overline{U}_-^+$ |
| CI | 7 | $U_{++}^+$ | $\overline{U}_{-+}^+$ | $U_{--}^+$ | $\overline{U}_{+-}^+$ |

## 2.2 Non-chiral class → chiral class

We construct the suspension isomorphism from a non-chiral class ($s_0 = 0$) to a chiral class ($s_0 = 1$). Let $\sigma_x, \sigma_y, \sigma_z$ be Pauli matrices, and define the Hamiltonian on the sphere $S^{d+D+1}$ as

$$\tilde{H}(\boldsymbol{k}, \boldsymbol{r}, \theta) = \cos\theta\, H(\boldsymbol{k}, \boldsymbol{r}) \otimes \sigma_z + \sin\theta\, \sigma_x, \quad \theta \in [0, \pi]. \tag{12}$$

At $\theta = 0, \pi$, the dependence on $\boldsymbol{k}, \boldsymbol{r}$ disappears, realizing the suspension $SS^{d+D} \cong S^{d+D+1}$. In what follows, the dependence on $\boldsymbol{k}, \boldsymbol{r}$ is not needed, so we omit these variables.

The Hamiltonian $\tilde{H}(\theta)$ has the chiral symmetry

$$\sigma_y \tilde{H}(\theta) \sigma_y = -\tilde{H}(\theta). \tag{13}$$

The mappings of the TRS or PHS symmetry $A$ and the additional symmetries $U_i$ are not unique; they are specified by $\mathbb{Z}_2$ values $\delta s_1, \delta t_{10}, \ldots, \delta t_{n0} \in \{0, 1\}$ within the freedom consistent with the definitions of symmetries in the chiral class (3), (5). We define

$$\tilde{A} = A \otimes (\sigma_z)^{1-s_1}(\sigma_x)^{\delta s_1}, \quad \delta s_1 \in \{0, 1\}, \tag{14}$$

$$\tilde{U}_i = U_i \otimes (i)^{t_{i0}\delta t_{i0}}(\sigma_z)^{t_{i0}}(\sigma_x)^{\delta t_{i0}}, \quad \delta t_{i0} \in \{0, 1\}, \quad i = 1, \ldots, n. \tag{15}$$

A straightforward computation yields the following symmetry relations and algebraic constraints:

$$\begin{cases} \tilde{A}\tilde{H}(\theta)\tilde{A}^{-1} = (-1)^{s_1+\delta s_1}\tilde{H}((-1)^{1-\delta s_1}\theta), \quad \tilde{A}^2 = (-1)^{s_2+(1-s_1)\delta s_1}, \\ \tilde{A}\sigma_y = (-1)^{s_1+\delta s_1}\sigma_y\tilde{A}, \\ \tilde{U}_i\tilde{H}(\theta)\tilde{U}_i^{-1} = (-1)^{t_{i0}+\delta t_{i0}}\tilde{H}((-1)^{\delta t_{i0}}\theta), \quad \tilde{U}_i^2 = 1, \\ \tilde{A}\tilde{U}_i = (-1)^{t_{i1}+(t_{i0}+\delta s_1)\delta t_{i0}+(1-s_1-\delta s_1)\delta t_{i0}}\tilde{U}_i\tilde{A}, \\ \sigma_y\tilde{U}_i = (-1)^{t_{i0}+\delta t_{i0}}\tilde{U}_i\sigma_y, \\ \tilde{U}_i\tilde{U}_j = (-1)^{u_{ij}+t_{i0}\delta t_{j0}+\delta t_{i0}t_{j0}}\tilde{U}_j\tilde{U}_i. \end{cases} \tag{16}$$

From this, we obtain the mapping of the labels of symmetry classes:

$$
\begin{cases}
s_0 = 0 \mapsto s_0 = 1, \\
s_1 \mapsto s_1 + \delta s_1, \\
s_2 \mapsto s_2 + (1 - s_1)\delta s_1, \\
t_{i0} \mapsto t_{i0} + \delta t_{i0}, \\
t_{i1} \mapsto t_{i1} + (t_{i0} + \delta s_1)\delta t_{i0}, \\
u_{ij} \mapsto u_{ij} + t_{i0}\delta t_{j0} + \delta t_{i0} t_{j0}.
\end{cases}
\tag{17}
$$

Equivalently, when $s$ is even, this can be written as

$$
\begin{cases}
s \mapsto s + (-1)^{\delta s_1}, \\
t_i \mapsto t_i + \delta t_{i0}(-1)^{\delta s_1}, \\
u_{ij} \mapsto u_{ij} + t_{i0}\delta t_{j0} + \delta t_{i0} t_{j0}.
\end{cases}
\tag{18}
$$

The presence or absence of inversion of the variable $\theta$ is specified by $1 - \delta s_1$ for $\tilde{A}$ and by $\delta t_{i0}$ for $\tilde{U}_i$. From the above, we obtain the following $K$-group isomorphism for even $s$:

$$
K_{\mathbb{R}+nU}\Big(s, \{t_i\}_i, \{u_{ij}\}_{i<j}; d, \{d_i\}_i, \{d_{ij}\}_{i<j}, \{d_{ijk}\}_{i<j<k}, \ldots; D, \{D_i\}_i, \{D_{ij}\}_{i<j}, \{D_{ijk}\}_{i<j<k}, \ldots\Big)
$$
$$
\xrightarrow{\cong} K_{\mathbb{R}+nU}\Big(s+1, \{t_i + \delta t_{i0}(-1)^{\delta s_1}\}_i, \{u_{ij} + t_{i0}\delta t_{j0} + \delta t_{i0} t_{j0}\}_{i<j};
$$
$$
d + (1 - \delta s_1), \{d_i + (1 - \delta s_1)\delta t_{i0}\}_i, \{d_{ij} + (1 - \delta s_1)\delta t_{i0}\delta t_{j0}\}_{i<j},
$$
$$
\{d_{ijk} + (1 - \delta s_1)\delta t_{i0}\delta t_{j0}\delta t_{k0}\}_{i<j<k}, \ldots;
$$
$$
D + \delta s_1, \{D_i + \delta s_1\delta t_{i0}\}_i, \{D_{ij} + \delta s_1\delta t_{i0}\delta t_{j0}\}_{i<j},
$$
$$
\{D_{ijk} + \delta s_1\delta t_{i0}\delta t_{j0}\delta t_{k0}\}_{i<j<k}, \ldots\Big).
\tag{19}
$$

As will be shown in the next subsection, this isomorphism also holds when $s$ is odd.

## 2.3 Chiral class → non-chiral class

In the same manner, we obtain a suspension isomorphism from the chiral class ($s_0 = 1$) to the non-chiral class ($s_0 = 0$). Define the Hamiltonian on $S^{d+D+1}$ by

$$
\tilde{H}(\theta) = \cos\theta\, H + \sin\theta\, \Gamma, \qquad \theta \in [0, \pi].
\tag{20}
$$

The mappings of the TRS or PHS operator $A$ and the additional symmetries $U_i$ are not unique; within the freedom compatible with the definitions of symmetries in the non-chiral class (2), (5), they are specified by $\mathbb{Z}_2$ values $\delta s_1, \delta t_{10}, \ldots, \delta t_{n0} \in \{0, 1\}$. We set

$$
\tilde{A} = \Gamma^{1-\delta s_1} A, \qquad \delta s_1 \in \{0, 1\},
\tag{21}
$$

$$
\tilde{U}_i = i^{\, t_{i0}\delta t_{i0}}\Gamma^{\delta t_{i0}} U_i, \qquad \delta t_{i0} \in \{0, 1\}, \quad i = 1, \ldots, n.
\tag{22}
$$

A straightforward computation yields

$$
\begin{cases}
\tilde{A}\tilde{H}(\theta)\tilde{A}^{-1} = (-1)^{s_1 + \delta s_1 + 1}\tilde{H}\big((-1)^{1-\delta s_1}\theta\big), \quad \tilde{A}^2 = (-1)^{s_2 + s_1(1-\delta s_1)}, \\
\tilde{U}_i\tilde{H}(\theta)\tilde{U}_i^{-1} = (-1)^{t_{i0} + \delta t_{i0}}\tilde{H}\big((-1)^{\delta t_{i0}}\theta\big), \quad \tilde{U}_i^2 = 1, \\
\tilde{A}\tilde{U}_i = (-1)^{t_{i1} + (t_{i0} + \delta s_1)\delta t_{i0} + (s_1 + \delta s_1)\delta t_{i0}}\tilde{U}_i\tilde{A}, \\
\tilde{U}_i\tilde{U}_j = (-1)^{u_{ij} + t_{i0}\delta t_{j0} + \delta t_{i0} t_{j0}}\tilde{U}_j\tilde{U}_i.
\end{cases}
\tag{23}
$$

Hence the labels map as

$$\begin{cases} s_0 = 1 \mapsto s_0 = 0, \\ s_1 \mapsto s_1 + (1 - \delta s_1), \\ s_2 \mapsto s_2 + s_1(1 - \delta s_1), \\ t_{i0} \mapsto t_{i0} + \delta t_{i0}, \\ t_{i1} \mapsto t_{i1} + (t_{i0} + \delta s_1)\delta t_{i0}, \\ u_{ij} \mapsto u_{ij} + t_{i0}\delta t_{j0} + \delta t_{i0}t_{j0}. \end{cases} \tag{24}$$

This is equivalent to (18) for even $s$. Therefore, we obtain the $K$-group isomorphism (19) for odd $s$ as well.

## 2.4 Reduction to zero dimension

From the isomorphism (19), the $K$-group reduces to that in zero dimension, i.e. $d = D = 0$:[1]

$$K_{\mathbb{R}+nU}\Big(s, \{t_i\}_i, \{u_{ij}\}_{i<j}; d, \{d_i\}_i, \{d_{ij}\}_{i<j}, \{d_{ijk}\}_{i<j<k}, \dots; D, \{D_i\}_i, \{D_{ij}\}_{i<j}, \{D_{ijk}\}_{i<j<k}, \dots\Big)$$
$$\cong K_{\mathbb{R}+nU}\Big(s - \delta, \{t_i - \delta_i\}_i, \{u_{ij} + \delta_{ij} + t_i\delta_j + t_j\delta_i\}_{i<j}\Big). \tag{25}$$

Here we introduced

$$\delta = d - D, \qquad \delta_i = d_i - D_i, \qquad \delta_{ij} = d_{ij} - D_{ij}. \tag{26}$$

In the zero-dimensional case as on the right-hand side of (25), we will omit $d, D$ from the notation. As in the cases with only internal symmetries [23] and with a single $\mathbb{Z}_2$ point-group symmetry [21], the $K$-group depends only on the defect dimension $(\delta - 1)$ and on the labels $\delta_i, \delta_{ij}$ that characterize how the $\mathbb{Z}_2^{\times n}$ point-group symmetry acts on the $\delta$-dimensional real space whose boundary is the defect.

Consequently, the $K$-group does not depend on the numbers of variables flipped in common by three or more generators, i.e. on $d_{ijk}, d_{ijkl}, \dots$ and $D_{ijk}, D_{ijkl}, \dots$. This follows from the fact that, in the isomorphism (19), the labels specifying the symmetry class $s, t_i, u_{ij}$ are insensitive to the product $\delta t_{i_1 0} \cdots \delta t_{i_r 0}$ with $r \geq 3$ indicating triple or higher overlaps. A minimal example exhibiting independence of $d_{ijk}$ is $d = 4$, $d_1 = d_2 = d_3 = 2$, $D = 0$, for which there are two possibilities $d_{123} = 0, 1$ but they give the same $K$-group.

In this paper, we do not explicitly compute the $K$-groups and periodic tables for three or more additional $\mathbb{Z}_2$ point-group symmetries. The computation is straightforward and proceeds in the same manner as in Sec. 3.1.

## 2.5 Complex AZ classes with unitary additional symmetries

For complex AZ classes (A, AIII) in the presence of additional $\mathbb{Z}_2^{\times n}$ point-group symmetries, the $K$-group isomorphism (25) also holds. In this setting, both the label $s$ specifying the complex AZ class and the labels $t_i$ specifying the additional symmetry types have period 2, i.e., $s, t_i \in \{0, 1\}$. Table 2 summarizes the relation between $(s, t_i)$ and the symmetry operators.

---

[1]To derive (25), it is convenient to use the transformation rule for the suspended variables $\tilde{u}_{ij} := u_{ij} + t_i t_j$, namely $\tilde{u}_{ij} \mapsto \tilde{u}_{ij} + \delta t_{i0}\delta t_{j0}$ following (18). Applying (19) a total of $d + D$ times, the symmetry class in 0D changes as $(\{t_i\}_i, \{\tilde{u}_{ij}\}_{i<j}) \mapsto (\{t_i - \delta_i\}_i, \{\tilde{u}_{ij} - \delta_{ij}\}_{i<j})$. Since $u_{ij} = \tilde{u}_{ij} - t_i t_j$, we get $u_{ij} \mapsto \tilde{u}_{ij} - \delta_{ij} - (t_i - \delta_i)(t_j - \delta_j)$, which yields (25).

Table 2: Complex AZ classes, additional symmetry classes, and the corresponding symmetry operators. The notation follows [21]. Here $U$ denotes a symmetry commuting with the Hamiltonian, while $\overline{U}$ denotes an antisymmetry anticommuting with the Hamiltonian. For the chiral class ($s = 1$), the subscript of $U$ indicates commutation/anticommutation with the chiral operator. Since there is no antiunitary symmetry, one can always redefine $U$ so that $U^2 = 1$, hence superscripts are omitted.

| AZ class | $s\backslash t_i$ | 0 | 1 |
|---|---|---|---|
| A | 0 | $U$ | $\overline{U}$ |
| AIII | 1 | $U_+$ | $\overline{U}_-$ |

## 2.6 Complex AZ classes with antiunitary additional symmetries

For complex AZ classes (A, AIII) with additional antiunitary $\mathbb{Z}_2^{\times n}$ point-group symmetries, the problem reduces to the case of real AZ classes with $\mathbb{Z}_2^{\times(n-1)}$ additional symmetries, in the same manner as in [21].

To avoid cumbersome notation, we present the case $n = 2$ for clarity. Since the product of two antiunitary symmetries is unitary, without loss of generality we take the first generator of $\mathbb{Z}_2^{\times n}$ to be antiunitary and the second to be unitary. We denote the antiunitary symmetry by $A$ and the unitary one by $U_1$. Combining the complex AZ class with the antiunitary symmetry $A$ yields eight symmetry classes, which we label by $s \in \{0, \ldots, 7\}$ with the binary representation $s = (s_2 s_1 s_0)_2$, in analogy with (1), (2), and (3). Let $P_A = \mathrm{diag}((-1)^{p_{A,1}}, \ldots, (-1)^{p_{A,d}})$ be the $\mathbb{Z}_2$ action of $A$ on momentum coordinates and $Q_A = \mathrm{diag}((-1)^{q_{A,1}}, \ldots, (-1)^{q_{A,D}})$ that on the defect (real-space–like) parameters. Taking into account the momentum inversion due to the antiunitarity of $A$, the symmetries of $H(\boldsymbol{k}, \boldsymbol{r})$ read

$$s_0 = 0 \text{ (non-chiral class)}: \quad AH(\boldsymbol{k}, \boldsymbol{r})A^{-1} = (-1)^{s_1}H(-P_A\boldsymbol{k}, Q_A\boldsymbol{r}), \qquad A^2 = (-1)^{s_2}, \quad (27)$$

$$s_0 = 1 \text{ (chiral class)}: \quad \begin{cases} AH(\boldsymbol{k}, \boldsymbol{r})A^{-1} = (-1)^{s_1}H(-P_A\boldsymbol{k}, Q_A\boldsymbol{r}), \qquad A^2 = (-1)^{s_2}, \\ \Gamma H(\boldsymbol{k}, \boldsymbol{r})\Gamma^{-1} = -H(\boldsymbol{k}, \boldsymbol{r}), \qquad \Gamma^2 = 1, \\ A\Gamma = (-1)^{s_1}\Gamma A. \end{cases} \quad (28)$$

For the additional unitary symmetry $U_1$ we introduce an additional class $t_1 \in \{0, 1, 2, 3\}$ and use the binary representation $t_1 = (t_{11} t_{10})_2$ to specify the algebraic relations, in analogy with (5).

Define the numbers of momenta flipped by $A$ and $U_1$ as

$$d_A = \sum_{\mu=1}^{d} p_{A,\mu}, \qquad d_1 = \sum_{\mu=1}^{d} p_{1,\mu}, \quad (29)$$

and similarly the numbers of defect parameters flipped by $A$ and $U_1$ as

$$D_A = \sum_{\mu=1}^{D} q_{A,\mu}, \qquad D_1 = \sum_{\mu=1}^{D} q_{1,\mu}. \quad (30)$$

The numbers of variables flipped simultaneously by $A$ and $U_1$ are

$$d_{A1} = \sum_{\mu=1}^{d} p_{A,\mu} p_{1,\mu}, \qquad D_{A1} = \sum_{\mu=1}^{D} q_{A,\mu} q_{1,\mu}. \quad (31)$$

We denote the $K$-group by

$$K_{\mathbb{C}+A+U}(s, t; d, d_A, d_1, d_{A1}; D, D_A, D_1, D_{A1}).\tag{32}$$

Now, the symmetry of $A$ in (27), (28) can be viewed as TRS or PHS of a real AZ class if we define the effective numbers of momentum-like and real-space–like variables flipped by $A$ as

$$\tilde{d} = \sum_{\mu=1}^{d}(1 - p_{A,\mu}) + \sum_{\mu=1}^{D}q_{A,\mu} = d - d_A + D_A,\tag{33}$$

$$\tilde{D} = \sum_{\mu=1}^{D}(1 - q_{A,\mu}) + \sum_{\mu=1}^{d}p_{A,\mu} = D - D_A + d_A,\tag{34}$$

as explained in [21]. With this effective partition $(\tilde{d}, \tilde{D})$, the numbers of variables flipped by $U_1$ become

$$\tilde{d}_1 = \sum_{\mu=1}^{d}(1 - p_{A,\mu})p_{1,\mu} + \sum_{\mu=1}^{D}q_{A,\mu}q_{1,\mu} = d_1 - d_{A1} + D_{A1},\tag{35}$$

$$\tilde{D}_1 = \sum_{\mu=1}^{D}(1 - q_{A,\mu})q_{1,\mu} + \sum_{\mu=1}^{d}p_{A,\mu}p_{1,\mu} = D_1 - D_{A1} + d_{A1}.\tag{36}$$

Therefore we obtain the isomorphisms

$$
\begin{aligned}
K_{\mathbb{C}+A+U}(s, t; d, d_A, d_1, d_{A1}; D, D_A, D_1, D_{A1}) &\cong K_{\mathbb{R}+U}(s, t; \tilde{d}, \tilde{d}_1; \tilde{D}, \tilde{D}_1)\\
&\cong K_{\mathbb{R}+U}(s - \tilde{d} + \tilde{D},\ t - \tilde{d}_1 + \tilde{D}_1)\\
&= K_{\mathbb{R}+U}(s - \delta + 2\delta_A,\ t - \delta_1 + 2\delta_{A1}),
\end{aligned}
\tag{37}
$$

where we set $\delta = d - D$, $\delta_A = d_A - D_A$, $\delta_1 = d_1 - D_1$, and $\delta_{A1} = d_{A1} - D_{A1}$. The groups $K_{\mathbb{R}+U}(s, t)$ are computed in [21] (see Table 3 below).

## 2.7 $K$-groups on tori

The isomorphism (25) holds when the $(d + D)$-dimensional parameter space is a sphere. For a torus, contributions from all sub-tori appear as direct summands.

For example, when the momentum space is a two-dimensional torus $T^2$ and the defect parameter space is a one-dimensional circle $S^1$, the total space is $T^2 \times S^1$. Denoting the coordinates by $k_1, k_2, r_1$, the following eight subspaces contribute as direct-sum components:

$$
\begin{cases}
S^3 : (k_1, k_2, r_1),\\
S^2 : (k_1, k_2),\\
S^2 : (k_1, r_1),\\
S^2 : (k_2, r_1),\\
S^1 : (k_1),\\
S^1 : (k_2),\\
S^1 : (r_1),\\
S^0.
\end{cases}
\tag{38}
$$

On each subspace, the $K$-group (11) is defined according to the numbers of variables flipped by the additional symmetries $U_i$, and the explicit classification groups are determined via the isomorphism (25).

Table 3: Classifying spaces in the presence of a single additional $\mathbb{Z}_2$ point-group symmetry of class $t$, in addition to a complex or real AZ class labeled by $s$ [21].

| Complex/Real | Additional symmetry class $t$ | Classifying space |
|---|---|---|
| Complex | 0 | $(C_s)^{\times 2}$ |
| | 1 | $C_{s+1}$ |
| Real | 0 | $(R_s)^{\times 2}$ |
| | 1 | $R_{s-1}$ |
| | 2 | $C_s$ |
| | 3 | $R_{s+1}$ |

# 3 Classification tables with additional $\mathbb{Z}_2^{\times 2}$ point group symmetry

In this section, the 0-dimensional $K$-groups are explicitly computed for the case with additional $\mathbb{Z}_2^{\times 2}$ point group symmetry, and the results are summarized in classification tables. For comparison, the case with a single additional $\mathbb{Z}_2$ symmetry was calculated in [21], and the corresponding results are collected in Table 3.

## 3.1 Computation of effective AZ classes in zero dimension

Due to the hierarchical structure, it is sufficient to determine the AZ classes for $s = 0$; the classifying spaces for arbitrary $s$ are then obtained by shifting the index accordingly. The procedure for computing AZ classes has already been established in [16]. In that approach, one first decomposes the unitary symmetry group into its irreducible representations and then applies the Wigner criteria, which provides a systematic calculation. Here, we instead perform the computation explicitly by a more elementary hand calculation.

For each set of additional symmetry classes $(t_1, t_2, u_{12})$, we introduce the following convenient notation. We denote the two unitary $\mathbb{Z}_2$ symmetries by $U_1$ and $U_2$. As in Table 1, we specify the signs of $(U_1)^2$ and $(U_2)^2$, their commutation relations with TRS and/or PHS, and whether they act as symmetries $U$ (commuting with the Hamiltonian) or antisymmetries $\overline{U}$ (anticommuting with the Hamiltonian). The mutual relation is encoded in the sign of $U_1 U_2 = (-1)^{u_{12}} U_2 U_1$ so that the algebraic data of the additional symmetries can be summarized by the triple

$$(U_1, U_2, (-1)^{u_{12}}). \tag{39}$$

For example, in the case $(t_1, t_2, u_{12}) = (1, 2, 0)$, the notation reads $(\overline{U}_1{}^+_-, U_2{}^+_-, +)$.

We now compute the AZ classes for $s = 0$. Note that the AZ class is invariant under the redefinitions

$$(U_1, U_2) \mapsto (U_2, U_1), \qquad (U_1, (i)^{u_{12}} U_1 U_2), \qquad ((i)^{u_{12}} U_1 U_2, U_2). \tag{40}$$

These redefinitions generate equivalences among the symmetry labels,

$$\begin{aligned}(t_1, t_2, u_{12}) &\sim (t_2, t_1, u_{12}) \sim (t_1, t_2 + t_1 + 2t_1 t_2 + 2u_{12}, u_{12}) \\ &\sim (t_1 + t_2 + 2t_1 t_2 + 2u_{12}, t_2, u_{12}).\end{aligned} \tag{41}$$

Hence, in the following, it is sufficient to restrict to the case $t_1 \le t_2$.

**Complex AZ classes with additional unitary $\mathbb{Z}_2^{\times 2}$ symmetries—**

- $(t_1, t_2, u_{12}) = (0, 0, 0) : (U_1, U_2, +)$.

  The system splits into four one-dimensional sectors labeled by $U_1 = \pm 1$, $U_2 = \pm 1$. Thus, $(\text{A})^{\times 4}$.

- $(t_1, t_2, u_{12}) = (0, 1, 0) : (U_1, \overline{U}_2, +)$, with $(0, 1, 0) \sim (1, 1, 0)$.

  The system decomposes into blocks with $U_1 = \pm 1$. Since $U_1 U_2 = U_2 U_1$, the operator $U_2$ acts as a chiral symmetry in each sector. Thus, $(\text{AIII})^{\times 2}$.

- $(t_1, t_2, u_{12}) = (0, 0, 1) : (U_1, U_2, -)$.

  Since $U_1 U_2 = -U_2 U_1$, the pair forms a nontrivial two-dimensional projective representation of $\mathbb{Z}_2 \times \mathbb{Z}_2$. Thus, A.

- $(t_1, t_2, u_{12}) = (0, 1, 1) : (U_1, \overline{U}_2, -)$, with $(0, 1, 1) \sim (1, 1, 1)$.

  Because $U_1 U_2 = -U_2 U_1$, the operator $U_2$ exchanges the sectors $U_1 = \pm 1$ and acts as an antisymmetry. Thus, A.

**Real AZ classes with additional unitary $\mathbb{Z}_2^{\times 2}$ symmetries—**

- $(t_1, t_2, u_{12}) = (0, 0, 0) : (U_{1+}^+, U_{2+}^+, +)$.

  In the four one-dimensional sectors $U_1 = \pm 1$, $U_2 = \pm 1$, TRS $T$ closes. Thus, $(\text{AI})^{\times 4}$.

- $(t_1, t_2, u_{12}) = (0, 1, 0) : (U_{1+}^+, \overline{U}_{2-}^+, +)$, with $(0, 1, 0) \sim (1, 1, 0)$.

  Since $U_1$ commutes with both $U_2$ and $T$, the system decomposes into blocks with $U_1 = \pm 1$. In each sector, TRS with $T^2 = 1$ exists, together with a PHS $(U_2 T)$ satisfying $(U_2 T)^2 = -1$. Thus, $(\text{CI})^{\times 2}$.

- $(t_1, t_2, u_{12}) = (0, 2, 0) : (U_{1+}^+, U_{2-}^+, +)$, with $(0, 2, 0) \sim (2, 2, 0)$.

  Since $U_1$ commutes with $U_2$ and $T$, the system splits into blocks with $U_1 = \pm 1$. Because $T U_2 = -U_2 T$, TRS $T$ exchanges the sectors $U_2 = \pm 1$. Thus, $(\text{A})^{\times 2}$.

- $(t_1, t_2, u_{12}) = (0, 3, 0) : (U_{1+}^+, \overline{U}_{2+}^+, +)$, with $(0, 3, 0) \sim (3, 3, 0)$.

  Since $U_1$ commutes with $U_2$ and $T$, the system splits into blocks with $U_1 = \pm 1$. In each sector, TRS with $T^2 = 1$ and PHS with $(U_2 T)^2 = 1$ are present. Thus, $(\text{BDI})^{\times 2}$.

- $(t_1, t_2, u_{12}) = (1, 2, 0) : (\overline{U}_{1-}^+, U_{2-}^+, +)$, with $(1, 2, 0) \sim (1, 3, 0) \sim (2, 3, 0)$.

  Since $T U_2 = -U_2 T$, TRS $T$ exchanges the sectors $U_2 = \pm 1$. Because $U_1 U_2 = U_2 U_1$, the operator $U_1$ acts as a chiral symmetry within each $U_2$ sector. Thus, AIII.

- $(t_1, t_2, u_{12}) = (0, 0, 1) : (U_{1+}^+, U_{2+}^+, -)$, with $(0, 0, 1) \sim (0, 2, 1)$.

  Since $T U_1 = U_1 T$, TRS $T$ closes in the sectors $U_1 = \pm 1$. Because $U_1 U_2 = -U_2 U_1$, the operator $U_2$ exchanges the sectors $U_1 = \pm 1$. Thus, AI.

- $(t_1, t_2, u_{12}) = (0, 1, 1) : (U_{1+}^+, \overline{U}_{2-}^+, -)$, with $(0, 1, 1) \sim (0, 3, 1) \sim (1, 3, 1)$.

  Since $T U_1 = U_1 T$, TRS $T$ closes in the sectors $U_1 = \pm 1$. Because $U_1 U_2 = -U_2 U_1$, the operator $U_2$ exchanges the sectors $U_1 = \pm 1$ as an antisymmetry. Thus, AI.

- $(t_1, t_2, u_{12}) = (1, 2, 1) : (\overline{U}_{1-}^+, U_{2-}^+, -)$, with $(1, 2, 1) \sim (1, 1, 1)$.

  Since $T U_2 = -U_2 T$, TRS $T$ exchanges the sectors $U_2 = \pm 1$. Because $U_1 T U_2 = U_2 U_1 T$, the operator $U_1 T$ closes within each $U_2$ sector and acts as a PHS with $(U_1 T)^2 = -1$. Thus, C.

Table 4: Classifying spaces in the presence of additional $\mathbb{Z}_2^{\times 2}$ symmetries specified by $(t_1, t_2, u_{12})$, on top of an AZ class labeled by $s$. Since the results are invariant under exchanging $t_1 \leftrightarrow t_2$, only the cases with $t_1 \leq t_2$ are shown.

| Complex/Real | Additional symmetry class $(t_1, t_2, u_{12})$ | AZ class for $s = 0$ | Classifying space |
|---|---|---|---|
| | $(0,0,0)$ | $(A)^{\times 4}$ | $(C_s)^{\times 4}$ |
| Complex | $(0,1,0) \sim (1,1,0)$ | $(AIII)^{\times 2}$ | $(C_{s+1})^{\times 2}$ |
| | $(0,0,1), (0,1,1) \sim (1,1,1)$ | A | $C_s$ |
| | $(0,0,0)$ | $(AI)^{\times 4}$ | $(R_s)^{\times 4}$ |
| | $(0,1,0) \sim (1,1,0)$ | $(CI)^{\times 2}$ | $(R_{s-1})^{\times 2}$ |
| | $(0,2,0) \sim (2,2,0)$ | $(A)^{\times 2}$ | $(C_s)^{\times 2}$ |
| | $(0,3,0) \sim (3,3,0)$ | $(BDI)^{\times 2}$ | $(R_{s+1})^{\times 2}$ |
| Real | $(1,2,0) \sim (1,3,0) \sim (2,3,0)$ | AIII | $C_{s+1}$ |
| | $(0,0,1) \sim (0,2,1), (0,1,1) \sim (0,3,1) \sim (1,3,1)$ | AI | $R_s$ |
| | $(1,1,1) \sim (1,2,1)$ | C | $R_{s-2}$ |
| | $(2,2,1)$ | AII | $R_{s+4}$ |
| | $(2,3,1) \sim (3,3,1)$ | D | $R_{s+2}$ |

- $(t_1, t_2, u_{12}) = (2,2,1) : (U_{1-}^+, U_{2-}^+, -)$.

  Since $T U_1 = -U_1 T$, TRS $T$ exchanges the sectors $U_1 = \pm 1$. Because $U_2 T U_1 = U_1 U_2 T$, the operator $U_2 T$ acts as a TRS with $(U_2 T)^2 = -1$. Thus, AII.

- $(t_1, t_2, u_{12}) = (2,3,1) : (U_{1-}^+, \overline{U}_{2+}^+, -)$, with $(2,3,1) \sim (3,3,1)$.

  Since $T U_1 = -U_1 T$, TRS $T$ exchanges the sectors $U_1 = \pm 1$. Because $U_2 T U_1 = U_1 U_2 T$, the operator $U_2 T$ closes within each $U_1$ sector and acts as a PHS with $(U_2 T)^2 = 1$. Thus, D.

From the above analysis, we have obtained the effective AZ classes realized for each additional symmetry class at $s = 0$. For a general $s$, the classifying space is obtained by shifting the AZ index by $s$, that is, by taking $s + s_0$ where $s_0$ corresponds to the effective AZ class. The results are summarized in Table 4.

The Abelian groups indicating the number of connected components of each classifying space (denoted by $\pi_0$) are listed in Table 5. From these results, the complete classification table is obtained for the case of real AZ classes with two additional unitary $\mathbb{Z}_2$ symmetries. In the following sections, we classify the cases according to the number of variables flipped by $U_1$ and $U_2$, and present the explicit classification tables.

## 3.2 Periodic tables for additional unitary $\mathbb{Z}_2^{\times 2}$ point-group symmetry

In this subsection, we present classification tables for the case where, in addition to complex or real AZ symmetries, an additional unitary $\mathbb{Z}_2^{\times 2}$ point-group symmetry is present. The $K$-group isomorphism is given by

$$
\begin{aligned}
& K_{\mathbb{C}/\mathbb{R}+2U}(s, t_1, t_2, u_{12}; d, d_1, d_2, d_{12}; D, D_1, D_2, D_{12}) \\
& \cong K_{\mathbb{C}/\mathbb{R}+2U}(s - \delta, t_1 - \delta_1, t_2 - \delta_2, u_{12} + \delta_{12} + t_1 \delta_2 + t_2 \delta_1).
\end{aligned}
\tag{42}
$$

Although in principle all possibilities of $(\delta_1, \delta_2, \delta_{12})$ may be considered, only the seven cases listed below are physically realized in spatial dimensions up to three.

- $\mathbb{Z}_2$ onsite + $\mathbb{Z}_2$ onsite.

  This is the case where $\delta_1 = \delta_2 = 0$.

- $\mathbb{Z}_2$ onsite + reflection.

  This is the case where $\delta_1 = 0, \delta_2 = 1$.

- $\mathbb{Z}_2$ onsite + $C_2$ rotation.

  This is the case where $\delta_1 = 0, \delta_2 = 2$.

- $\mathbb{Z}_2$ onsite + inversion.

  This is the case where $\delta_1 = 0, \delta_2 = 3$.

- Reflection + reflection (with a different reflection plane).

  This is the case where $\delta_1 = 1, \delta_2 = 1$, and $\delta_{12} = 0$. Note that the two reflections have different reflection planes; otherwise, the redefinition of $U_1$ as $U_1 U_2$ is recast as a $\mathbb{Z}_2$ on-site and reflection. This case is possible when the defect dimensions are larger than two.

- Reflection + $C_2$ rotation where the reflection plane is perpendicular to the rotation axis.

  This is the case where $\delta_1 = 1, \delta_2 = 2$, and $\delta_{12} = 0$. Note that when the rotation axis is parallel to the reflection plane ($\delta_{12} = 1$), then the redefinition of $U_2$ as $U_1 U_2$ is recast as two reflections with $\delta_{12} = 0$. This is only possible for bulk classification in three dimensions.

- $C_2$ rotation + $C_2$ rotation (with the rotation axes perpendicular to each other).

  This is the case where $\delta_1 = 2, \delta_2 = 2$, and $\delta_{12} = 1$. This is only possible for bulk classification in three dimensions.

The classification tables depend only on the four parameters obtained from the isomorphism (42),

$$\tilde{s} = s - \delta, \quad \tilde{t}_1 = t_1 - \delta_1, \quad \tilde{t}_2 = t_2 - \delta_2, \quad \tilde{u}_{12} = u_{12} + \delta_{12} + t_1 \delta_2 + t_2 \delta_1. \tag{43}$$

Therefore, it is sufficient to provide classification tables for the twelve cases of $(\tilde{t}_1, \tilde{t}_2, \tilde{u}_{12})$ listed in Table 4. The results are summarized in Table 6.

Table 5: AZ classes and classifying spaces. The periodicity of $s$ is 2 for complex AZ classes and 8 for real AZ classes. $\pi_0$ denotes the number of connected components.

| AZ class | $s$ | Classifying space | $\pi_0$ |
|---|---|---|---|
| A | 0 | $C_0$ | $\mathbb{Z}$ |
| AIII | 1 | $C_1$ | 0 |
| AI | 0 | $R_0$ | $\mathbb{Z}$ |
| BDI | 1 | $R_1$ | $\mathbb{Z}_2$ |
| D | 2 | $R_2$ | $\mathbb{Z}_2$ |
| DIII | 3 | $R_3$ | 0 |
| AII | 4 | $R_4$ | $\mathbb{Z}$ |
| CII | 5 | $R_5$ | 0 |
| C | 6 | $R_6$ | 0 |
| CI | 7 | $R_7$ | 0 |

Table 6: Periodic table for unitary $\mathbb{Z}_2^{\times 2}$ point-group symmetry. Here, $\delta_1$ and $\delta_2$ denote the numbers of variables flipped by $U_1$ and $U_2$, respectively, while $\delta_{12}$ denotes the number of variables flipped simultaneously by both $U_1$ and $U_2$. For each AZ class, the additional symmetry classes $t_1, t_2$ associated with $U_1, U_2$ are determined by Tables 1 and 2. The parameter $u_{12} \in \{0, 1\}$ specifies whether $U_1$ and $U_2$ commute or anticommute. Given the data $\delta_1, \delta_2, \delta_{12}, t_1, t_2, u_{12}$ of the unitary $\mathbb{Z}_2^{\times 2}$ point-group symmetry, one computes the triple $(t_1 - \delta_1, t_2 - \delta_2, u_{12} + \delta_{12} + t_1\delta_2 + t_2\delta_1)$, and the corresponding classification result is obtained from the relevant row in the first column.

| $(t_1 - \delta_1, t_2 - \delta_2, u_{12} + \delta_{12} + t_1\delta_2 + t_2\delta_1)$ | Classifying space | AZ class | $\delta = 0$ | $\delta = 1$ | $\delta = 2$ | $\delta = 3$ | $\delta = 4$ | $\delta = 5$ | $\delta = 6$ | $\delta = 7$ |
|---|---|---|---|---|---|---|---|---|---|---|
| $(0,0,0)$ | $(C_{s-\delta})^4$ | A | $\mathbb{Z}^4$ | 0 | $\mathbb{Z}^4$ | 0 | $\mathbb{Z}^4$ | 0 | $\mathbb{Z}^4$ | 0 |
| | | AIII | 0 | $\mathbb{Z}^4$ | 0 | $\mathbb{Z}^4$ | 0 | $\mathbb{Z}^4$ | 0 | $\mathbb{Z}^4$ |
| $(1,0,0) \sim (0,1,0) \sim (1,1,0)$ | $(C_{s+1-\delta})^2$ | A | 0 | $\mathbb{Z}^2$ | 0 | $\mathbb{Z}^2$ | 0 | $\mathbb{Z}^2$ | 0 | $\mathbb{Z}^2$ |
| | | AIII | $\mathbb{Z}^2$ | 0 | $\mathbb{Z}^2$ | 0 | $\mathbb{Z}^2$ | 0 | $\mathbb{Z}^2$ | 0 |
| $(0,0,1),$ $(1,0,1) \sim (0,1,1) \sim (1,1,1)$ | $C_{s-\delta}$ | A | $\mathbb{Z}$ | 0 | $\mathbb{Z}$ | 0 | $\mathbb{Z}$ | 0 | $\mathbb{Z}$ | 0 |
| | | AIII | 0 | $\mathbb{Z}$ | 0 | $\mathbb{Z}$ | 0 | $\mathbb{Z}$ | 0 | $\mathbb{Z}$ |
| $(0,0,0)$ | $(R_{s-\delta})^4$ | AI | $\mathbb{Z}^4$ | 0 | 0 | 0 | $(2\mathbb{Z})^4$ | 0 | $\mathbb{Z}_2^4$ | $\mathbb{Z}_2^4$ |
| | | BDI | $\mathbb{Z}_2^4$ | $\mathbb{Z}^4$ | 0 | 0 | 0 | $(2\mathbb{Z})^4$ | 0 | $\mathbb{Z}_2^4$ |
| | | D | $\mathbb{Z}_2^4$ | $\mathbb{Z}_2^4$ | $\mathbb{Z}^4$ | 0 | 0 | 0 | $(2\mathbb{Z})^4$ | 0 |
| | | DIII | 0 | $\mathbb{Z}_2^4$ | $\mathbb{Z}_2^4$ | $\mathbb{Z}^4$ | 0 | 0 | 0 | $(2\mathbb{Z})^4$ |
| | | AII | $(2\mathbb{Z})^4$ | 0 | $\mathbb{Z}_2^4$ | $\mathbb{Z}_2^4$ | $\mathbb{Z}^4$ | 0 | 0 | 0 |
| | | CII | 0 | $(2\mathbb{Z})^4$ | 0 | $\mathbb{Z}_2^4$ | $\mathbb{Z}_2^4$ | $\mathbb{Z}^4$ | 0 | 0 |
| | | C | 0 | 0 | $(2\mathbb{Z})^4$ | 0 | $\mathbb{Z}_2^4$ | $\mathbb{Z}_2^4$ | $\mathbb{Z}^4$ | 0 |
| | | CI | 0 | 0 | 0 | $(2\mathbb{Z})^4$ | 0 | $\mathbb{Z}_2^4$ | $\mathbb{Z}_2^4$ | $\mathbb{Z}^4$ |
| $(1,0,0) \sim (0,1,0) \sim (1,1,0)$ | $(R_{s-1-\delta})^2$ | AI | 0 | 0 | 0 | $(2\mathbb{Z})^2$ | 0 | $\mathbb{Z}_2^2$ | $\mathbb{Z}_2^2$ | $\mathbb{Z}^2$ |
| | | BDI | $\mathbb{Z}^2$ | 0 | 0 | 0 | $(2\mathbb{Z})^2$ | 0 | $\mathbb{Z}_2^2$ | $\mathbb{Z}_2^2$ |
| | | D | $\mathbb{Z}_2^2$ | $\mathbb{Z}^2$ | 0 | 0 | 0 | $(2\mathbb{Z})^2$ | 0 | $\mathbb{Z}_2^2$ |
| | | DIII | $\mathbb{Z}_2^2$ | $\mathbb{Z}_2^2$ | $\mathbb{Z}^2$ | 0 | 0 | 0 | $(2\mathbb{Z})^2$ | 0 |
| | | AII | 0 | $\mathbb{Z}_2^2$ | $\mathbb{Z}_2^2$ | $\mathbb{Z}^2$ | 0 | 0 | 0 | $(2\mathbb{Z})^2$ |
| | | CII | $(2\mathbb{Z})^2$ | 0 | $\mathbb{Z}_2^2$ | $\mathbb{Z}_2^2$ | $\mathbb{Z}^2$ | 0 | 0 | 0 |
| | | C | 0 | $(2\mathbb{Z})^2$ | 0 | $\mathbb{Z}_2^2$ | $\mathbb{Z}_2^2$ | $\mathbb{Z}^2$ | 0 | 0 |
| | | CI | 0 | 0 | $(2\mathbb{Z})^2$ | 0 | $\mathbb{Z}_2^2$ | $\mathbb{Z}_2^2$ | $\mathbb{Z}^2$ | 0 |
| $(2,0,0) \sim (0,2,0) \sim (2,2,0)$ | $(C_{s-\delta})^2$ | AI,D,AII,C | $\mathbb{Z}^2$ | 0 | $\mathbb{Z}^2$ | 0 | $\mathbb{Z}^2$ | 0 | $\mathbb{Z}^2$ | 0 |
| | | BDI,DIII,CII,CI | 0 | $\mathbb{Z}^2$ | 0 | $\mathbb{Z}^2$ | 0 | $\mathbb{Z}^2$ | 0 | $\mathbb{Z}^2$ |
| $(3,0,0) \sim (0,3,0) \sim (3,3,0)$ | $(R_{s+1-\delta})^2$ | AI | $\mathbb{Z}_2^2$ | $\mathbb{Z}^2$ | 0 | 0 | 0 | $(2\mathbb{Z})^2$ | 0 | $\mathbb{Z}_2^2$ |
| | | BDI | $\mathbb{Z}_2^2$ | $\mathbb{Z}_2^2$ | $\mathbb{Z}^2$ | 0 | 0 | 0 | $(2\mathbb{Z})^2$ | 0 |
| | | D | 0 | $\mathbb{Z}_2^2$ | $\mathbb{Z}_2^2$ | $\mathbb{Z}^2$ | 0 | 0 | 0 | $(2\mathbb{Z})^2$ |
| | | DIII | $(2\mathbb{Z})^2$ | 0 | $\mathbb{Z}_2^2$ | $\mathbb{Z}_2^2$ | $\mathbb{Z}^2$ | 0 | 0 | 0 |
| | | AII | 0 | $(2\mathbb{Z})^2$ | 0 | $\mathbb{Z}_2^2$ | $\mathbb{Z}_2^2$ | $\mathbb{Z}^2$ | 0 | 0 |
| | | CII | 0 | 0 | $(2\mathbb{Z})^2$ | 0 | $\mathbb{Z}_2^2$ | $\mathbb{Z}_2^2$ | $\mathbb{Z}^2$ | 0 |
| | | C | 0 | 0 | 0 | $(2\mathbb{Z})^2$ | 0 | $\mathbb{Z}_2^2$ | $\mathbb{Z}_2^2$ | $\mathbb{Z}^2$ |
| | | CI | $\mathbb{Z}^2$ | 0 | 0 | 0 | $(2\mathbb{Z})^2$ | 0 | $\mathbb{Z}_2^2$ | $\mathbb{Z}_2^2$ |

Continuation of Table 6.

| $(t_1-\delta_1,t_2-\delta_2,u_{12}+\delta_{12}+t_1\delta_2+t_2\delta_1)$ | Classifying space | AZ class | $\delta=0$ | $\delta=1$ | $\delta=2$ | $\delta=3$ | $\delta=4$ | $\delta=5$ | $\delta=6$ | $\delta=7$ |
|---|---|---|---|---|---|---|---|---|---|---|
| $(2,1,0)\sim(1,2,0)\sim(3,1,0)$ | $C_{s+1-\delta}$ | AI,D,AII,C | 0 | $\mathbb{Z}$ | 0 | $\mathbb{Z}$ | 0 | $\mathbb{Z}$ | 0 | $\mathbb{Z}$ |
| $\sim(1,3,0)\sim(3,2,0)\sim(2,3,0)$ | | BDI,DIII,CII,CI | $\mathbb{Z}$ | 0 | $\mathbb{Z}$ | 0 | $\mathbb{Z}$ | 0 | $\mathbb{Z}$ | 0 |
| $(0,0,1)\sim(2,0,1)\sim(0,2,1),$ | $R_{s-\delta}$ | AI | $\mathbb{Z}$ | 0 | 0 | 0 | $2\mathbb{Z}$ | 0 | $\mathbb{Z}_2$ | $\mathbb{Z}_2$ |
| | | BDI | $\mathbb{Z}_2$ | $\mathbb{Z}$ | 0 | 0 | 0 | $2\mathbb{Z}$ | 0 | $\mathbb{Z}_2$ |
| | | D | $\mathbb{Z}_2$ | $\mathbb{Z}_2$ | $\mathbb{Z}$ | 0 | 0 | 0 | $2\mathbb{Z}$ | 0 |
| $(1,0,1)\sim(0,1,1)\sim(3,0,1)$ | | DIII | 0 | $\mathbb{Z}_2$ | $\mathbb{Z}_2$ | $\mathbb{Z}$ | 0 | 0 | 0 | $2\mathbb{Z}$ |
| $\sim(0,3,1)\sim(3,1,1)\sim(1,3,1)$ | | AII | $2\mathbb{Z}$ | 0 | $\mathbb{Z}_2$ | $\mathbb{Z}_2$ | $\mathbb{Z}$ | 0 | 0 | 0 |
| | | CII | 0 | $2\mathbb{Z}$ | 0 | $\mathbb{Z}_2$ | $\mathbb{Z}_2$ | $\mathbb{Z}$ | 0 | 0 |
| | | C | 0 | 0 | $2\mathbb{Z}$ | 0 | $\mathbb{Z}_2$ | $\mathbb{Z}_2$ | $\mathbb{Z}$ | 0 |
| | | CI | 0 | 0 | 0 | $2\mathbb{Z}$ | 0 | $\mathbb{Z}_2$ | $\mathbb{Z}_2$ | $\mathbb{Z}$ |
| $(1,1,1)\sim(2,1,1)\sim(1,2,1)$ | $R_{s-2-\delta}$ | AI | 0 | 0 | $2\mathbb{Z}$ | 0 | $\mathbb{Z}_2$ | $\mathbb{Z}_2$ | $\mathbb{Z}$ | 0 |
| | | BDI | 0 | 0 | 0 | $2\mathbb{Z}$ | 0 | $\mathbb{Z}_2$ | $\mathbb{Z}_2$ | $\mathbb{Z}$ |
| | | D | $\mathbb{Z}$ | 0 | 0 | 0 | $2\mathbb{Z}$ | 0 | $\mathbb{Z}_2$ | $\mathbb{Z}_2$ |
| | | DIII | $\mathbb{Z}_2$ | $\mathbb{Z}$ | 0 | 0 | 0 | $2\mathbb{Z}$ | 0 | $\mathbb{Z}_2$ |
| | | AII | $\mathbb{Z}_2$ | $\mathbb{Z}_2$ | $\mathbb{Z}$ | 0 | 0 | 0 | $2\mathbb{Z}$ | 0 |
| | | CII | 0 | $\mathbb{Z}_2$ | $\mathbb{Z}_2$ | $\mathbb{Z}$ | 0 | 0 | 0 | $2\mathbb{Z}$ |
| | | C | $2\mathbb{Z}$ | 0 | $\mathbb{Z}_2$ | $\mathbb{Z}_2$ | $\mathbb{Z}$ | 0 | 0 | 0 |
| | | CI | 0 | $2\mathbb{Z}$ | 0 | $\mathbb{Z}_2$ | $\mathbb{Z}_2$ | $\mathbb{Z}$ | 0 | 0 |
| $(2,2,1)$ | $R_{s+4-\delta}$ | AI | $2\mathbb{Z}$ | 0 | $\mathbb{Z}_2$ | $\mathbb{Z}_2$ | $\mathbb{Z}$ | 0 | 0 | 0 |
| | | BDI | 0 | $2\mathbb{Z}$ | 0 | $\mathbb{Z}_2$ | $\mathbb{Z}_2$ | $\mathbb{Z}$ | 0 | 0 |
| | | D | 0 | 0 | $2\mathbb{Z}$ | 0 | $\mathbb{Z}_2$ | $\mathbb{Z}_2$ | $\mathbb{Z}$ | 0 |
| | | DIII | 0 | 0 | 0 | $2\mathbb{Z}$ | 0 | $\mathbb{Z}_2$ | $\mathbb{Z}_2$ | $\mathbb{Z}$ |
| | | AII | $\mathbb{Z}$ | 0 | 0 | 0 | $2\mathbb{Z}$ | 0 | $\mathbb{Z}_2$ | $\mathbb{Z}_2$ |
| | | CII | $\mathbb{Z}_2$ | $\mathbb{Z}$ | 0 | 0 | 0 | $2\mathbb{Z}$ | 0 | $\mathbb{Z}_2$ |
| | | C | $\mathbb{Z}_2$ | $\mathbb{Z}_2$ | $\mathbb{Z}$ | 0 | 0 | 0 | $2\mathbb{Z}$ | 0 |
| | | CI | 0 | $\mathbb{Z}_2$ | $\mathbb{Z}_2$ | $\mathbb{Z}$ | 0 | 0 | 0 | $2\mathbb{Z}$ |
| $(3,2,1)\sim(2,3,1)\sim(3,3,1)$ | $R_{s+2-\delta}$ | AI | $\mathbb{Z}_2$ | $\mathbb{Z}_2$ | $\mathbb{Z}$ | 0 | 0 | 0 | $2\mathbb{Z}$ | 0 |
| | | BDI | 0 | $\mathbb{Z}_2$ | $\mathbb{Z}_2$ | $\mathbb{Z}$ | 0 | 0 | 0 | $2\mathbb{Z}$ |
| | | D | $2\mathbb{Z}$ | 0 | $\mathbb{Z}_2$ | $\mathbb{Z}_2$ | $\mathbb{Z}$ | 0 | 0 | 0 |
| | | DIII | 0 | $2\mathbb{Z}$ | 0 | $\mathbb{Z}_2$ | $\mathbb{Z}_2$ | $\mathbb{Z}$ | 0 | 0 |
| | | AII | 0 | 0 | $2\mathbb{Z}$ | 0 | $\mathbb{Z}_2$ | $\mathbb{Z}_2$ | $\mathbb{Z}$ | 0 |
| | | CII | 0 | 0 | 0 | $2\mathbb{Z}$ | 0 | $\mathbb{Z}_2$ | $\mathbb{Z}_2$ | $\mathbb{Z}$ |
| | | C | $\mathbb{Z}$ | 0 | 0 | 0 | $2\mathbb{Z}$ | 0 | $\mathbb{Z}_2$ | $\mathbb{Z}_2$ |
| | | CI | $\mathbb{Z}_2$ | $\mathbb{Z}$ | 0 | 0 | 0 | $2\mathbb{Z}$ | 0 | $\mathbb{Z}_2$ |



# 4  Conclusion

In this paper, we have systematically studied the classification of topological insulators and superconductors with multiple additional $\mathbb{Z}_2$ point-group symmetries. By employing suspension isomorphisms, we showed that higher-dimensional classification groups can be reduced to lower-dimensional data, thereby revealing a hierarchical structure of the classification. Furthermore, for both real and complex AZ classes, we derived a general classification formula in the presence of an arbitrary number of unitary $\mathbb{Z}_2$ point-group symmetries, and demonstrated that the classification groups are uniquely determined by a finite set of $\mathbb{Z}_4$ and $\mathbb{Z}_2$ parameters. As a concrete example, we carried out detailed computations for the case of $\mathbb{Z}_2^{\times 2}$ point-group symmetry, and confirmed the effectiveness of the method by presenting the complete classification tables. These results contribute to a systematic understanding of higher-order topological insulators and superconductors realized under multiple $\mathbb{Z}_2$ point-group symmetries, and serve as a practical framework for explicit classification in the presence of crystalline symmetries.

# Acknowledgments

**Funding information**   We were supported by JST CREST Grant No. JPMJCR19T2, and JSPS KAKENHI Grant No. JP22H05118 and JP23H01097.

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
