# Peer review of "Classification of topological insulators and superconductors with multiple order-two point group symmetries"

_SciPost Physics Core, doi:SciPost Phys. Core 8, 088 (2025)_

## Round 1 · Referee Report · Anonymous (Referee 1) · 2025-10-31

Strengths
1) Complete and completely general solution of the classification problem of topological insulators and superconductors with multiple order-two crystalline symmetries. Until now, general classifications (which apply to all tenfold-way symmetry classes) exist for the case of a single order-two crystalline symmetry only.
2) Extremely concise manuscript, which makes it very efficient to read for expert readers.
Weaknesses
1) The manuscript is so concise, that a non-expert reader has no chance.
Report
The orginal 2014 classification of topological insulators and superconductors with a single order-two symmetry by Shiozaki and Sato used a method of compelling elegance: Building on earlier work by Teo and Kane, an isomorphism was constructed that related the classification in d dimensions to that in d-1 dimensions. Recursively applying this construction, the full classification could be obtained from that in zero dimensions. The classification problem in zero dimensions is simple enough, that it can easily be solved by explicit calculation.
In the present manuscript, Shiozaki now shows that the same strategy can be applied to classify topological insulators and superconductors with multiple order-two crystalline symmetries. The generalization of his previous work with Sato to the case of multiple symmetries proves to be suprisingly straightforward, at least once an appropriate notation has been introduced to deal with the various possibilities for symmetries, antisymmetries, antiunitary symmetries, and antiunitary antisymmetries that appear in this setting. The manuscript describes the general strategy to map the classification in d dimensions to that in zero dimensions and it gives the explicit calculation in zero dimensions for the case of two crystalline symmetries. For researchers interested in topological classifications, I find this an interesting and important development that deserves to be published.
For the expert reader who is familiar with the original 2014 article by Shiozaki and Sato the manuscript is an efficient read, because Shiozaki refrains from giving a lengthy introduction or an explanation of the details behind the classification strategy. The manuscript contains precisely the right amount of additional information to understand how the classification procedure of the 2014 Shiozaki and Sato article can be extended to Bloch Hamiltonians with multiple order-two symmetries. The essential piece of information in the manuscript, the expressions for dimension reduction isomorphism, is cumbersome even for the expert reader, but this complexity is intrinsic to the problem at hand and one can decipher the expressions for a specific case one is interested in. (I have tried this for a few examples.)
I strongly doubt, however, that a reader without such a background can get anything out of the manuscript without prior intensive study of Ref. 21. To make the present article accessible to such a readership would basically require writing a completely new text, which is then likely to have a length that is two or three times the length of the present manuscript.
If SciPost is content publishing a concise article that reports an important result and contains all relevant information in a manner that is accessible to experts only, I see no reason not to publish the present manuscript in SciPost. Given the technical nature of the manuscript and the close connection to Ref. 21, I do not find publication in SciPost Core appropriate. If being self-contained is a requirement for articles published in SciPost, the present manuscript completely misses the mark and I do not see a route to publication in SciPost without requiring the author to basically write a completely new manuscript.
Recommendation
Accept in alternative Journal (see Report)

---

## Editorial Decision

published